
**ESD Ideas: Why are glacial inceptions slower than terminations?**
Christine Ramadhin[1,2], Chuixiang Yi[2,1]†
[1]Earth and Environmental Sciences Department, Graduate Center, City University of New York,
New York, NY 10016, USA
[2]School of Earth and Environmental Sciences, Queens College, City University of New York,
New York, NY 11367, USA.
*Correspondence to:* Chuixiang Yi (cyi@qc.cuny.edu)
**Abstract.** The Earth's climate during the Quaternary is dominated by short warm interglacials
and longer cold glaciations paced by external forcings such as changes in insolation. Although
not observed in the solar radiation changes, the time series of the cycles display asymmetry since
transitions to full glacial conditions are slower than termination of glaciations. Here an idea is
proposed for the slower transition by identifying and describing two negative sea ice feedbacks,
dominant during the glaciation process that could serve as a control of the intermediate stage and
decrease the pace of the process.
Paleoclimate data show that the Earth's climate of the last 2.6Ma is dominated by cold
glaciations lasting about a 100ka with extensive glaciers and warm interglacials with little global
ice cover lasting 10-30ka (Imbrie et al., 1992). The termination of a glacial period occurs rapidly
after a maximum in Northern Hemisphere (NH) summer insolation is crossed while glacial
inceptions are similarly triggered by astronomical forcing when a threshold of insolation
minimum is surpassed (Lisiecki, 2010). However, unlike terminations, which take thousands of
years, the changeover to a glacial period takes tens of thousands of years to be completed
resulting in an interesting asymmetrical shape for which there is yet no consensus on the
mechanism(s) (Tziperman and Gildor, 2003). There is a plethora of previous research which
have identified several feedback mechanisms to explain glacial-interglacial asymmetry, for
example, varying ice sheet volumes (Le Treut and Ghil, 1983).
While it is mostly agreed that astronomical forcings trigger glacial-interglacial transitions, a
similar shape is not observed in insolation changes suggesting a nonlinear response by the
climate system (Lisiecki and Raymo, 2007). Negative feedbacks work to maintain the stability of
an equilibrium state while positive feedbacks favor instability and regime transitions (Rial et al.,
2003). Therefore, it seems intuitive that negative feedbacks would play critical roles in slowing
the pace of a transition between equilibrium states. Here it is proposed that negative feedbacks
are responsible for the development of the intermediary stage observed during the glaciation and
two negative sea ice feedbacks are specified as the critical feedbacks.


This paper suggests that sea ice-precipitation is one of the first feedbacks to become dominant
during glaciations as **Figure 1** illustrates. The red arrow in the inset of **Figure 1** for each of the
four panels show the stage of the glacial inception process being addressed. Looking at the
Arctic region with interglacial conditions, there is little sea ice formation but strong energy and
mass ocean-atmosphere exchange (**Figure 1A**). The glaciation process is initiated after the
insolation minimum is crossed and NH temperature decreases (Lisiecki 2010). This allows sea
ice to extend rapidly, increasing albedo, further decreasing temperatures explaining the initial
drop observed in the temperature data shown in the inset of **Figure 1**. Arctic sea ice controls
climate by regulating albedo and air-sea exchange of both energy and gases (Weyl, 1968). Both
proxy and modeling data indicate significant ice cover over the Arctic Ocean during the last
glacial maximum (Colleoni et al., 2009; Jakobsson et al., 2016). The increased sea ice coverage
and decreased air-sea exchange may have caused cold, dry atmospheric conditions to develop,
reducing precipitation as depicted in **Figure 1B**. With less precipitation in the Arctic region,
there is less ice accumulation, and sea ice growth is hindered. Hydrogen isotope ratios (d-excess
values) confirm that sea ice controls regional Arctic precipitation by contributing to drier
conditions on Greenland as sea ice extent increases (Kopec et al., 2016).
The extensive sea ice formed now becomes starved as precipitation and ice accumulation
become limited, the newly formed sea ice thins and becomes more vulnerable to ablation.
Enhanced ocean movements can result in a rapid ablation as shown in **Figure 1C** and inset (red
arrow shows the stage of transition) and increased temperatures resulting in an intermediate stage
with higher temperature or mild glacial conditions as described by Paillard (1998). This stage
remains dominant allowing increased evaporation due to the newly uncovered ocean, and
increased precipitation. As overall temperatures remain cooler than during interglacial with low
insolation values prevailing, the broken sea ice reforms into a sturdier sheet that is more resistant
to ocean turn over (**Figure 1D**).
The second dominant feedback proposed is sea ice-insulation feedback as portrayed in **Figure 1**.
Results from the atmospheric HIRHAM regional climate model showed sea ice acts to regulate
air-sea heat flux by allowing stronger heat flux when sea ice thickness is reduced (Curry et al.,
1995). With the formation of sea-ice in the Arctic Ocean, as explained above and shown in
**Figure 1B**, and air-sea energy exchange reduced, there is reduced heat loss and geothermal
energy build up in the deep ocean increasing seawater buoyancy and at some critical point
leading to vertical ocean turbulence and sea ice ablation, illustrated in **Figure 1C**. In a study of
Dansgaard-Oescheger events, the ECBilt-Clio model demonstrates the effect of sea ice on deep
ocean temperature where a $13\text{-}15 \times 10^6 \text{km}^2$ sea ice extent led to the deep ocean temperature
increase of $2\text{-}4^\circ\text{C}$ at $1.5\text{-}3.5\text{km}$ depth, and $2\text{-}5^\circ\text{C}$ (Rial and Saha, 2011). This induces instability
with cold dense water at the surface and warmer water in the deep resulting in turbulent vertical
mixing at some point, and sea ice ablation. The first proposed feedback sea ice-precipitation
likely enhances this sea ice disintegration; where decreased precipitation results in thinning sea
ice and increased susceptibility to perturbations such as turbulent mixing.
Sea ice disintegration decreases albedo resulting in higher temperatures and the development of
the intermediary stage depicted in **Figure 1C**, where there is a return to almost interglacial
conditions. This stage remains dominant temporarily with increased evaporation allowed by the
newly uncovered ocean and increased precipitation. Overall, temperatures cooler than



interglacial temperatures prevail during the intermediate stage as insolation values remain low,
the broken sea ice reforms, shown in **Figure 1D**, and is much sturdier since not all the first-stage
ice was lost. The more durable sea ice has greater resilience to ocean turnover, and instead of
breaking up with ocean turbulence, there are changes in local North Atlantic convection sites.
Shifting the position of the North Atlantic Deep Water (NADW) formation is demonstrated by
previous studies, for example, Rahmstorf, (2006).
Paillard, (1998) showed the existence of an intermediate stage during glacial inception in which
an ice volume threshold must be crossed before the transition to full glacial conditions can occur
and help simulate the asymmetry of the glacial cycles. Building on this concept, two negative
feedbacks, sea ice-precipitation and sea ice-insulation that provide a physical cause of the
intermediary stage during glaciations that makes the process much slower than terminations are
presented. The dominance of these two feedbacks have implications for models that will
replicate the climate dynamics of the glacial-interglacial cycle transitions and similar critical
transitions of dynamical systems by emphasizing the role of negative feedbacks. Given the
unusually fast rate of anthropogenic changes the Earth system is currently undergoing, the risk of
crossing thresholds and transitioning to another climate state becomes greater. Therefore
improving understanding of how negative feedbacks facilitate climate regime change will help in
estimating the speed of transitions.

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





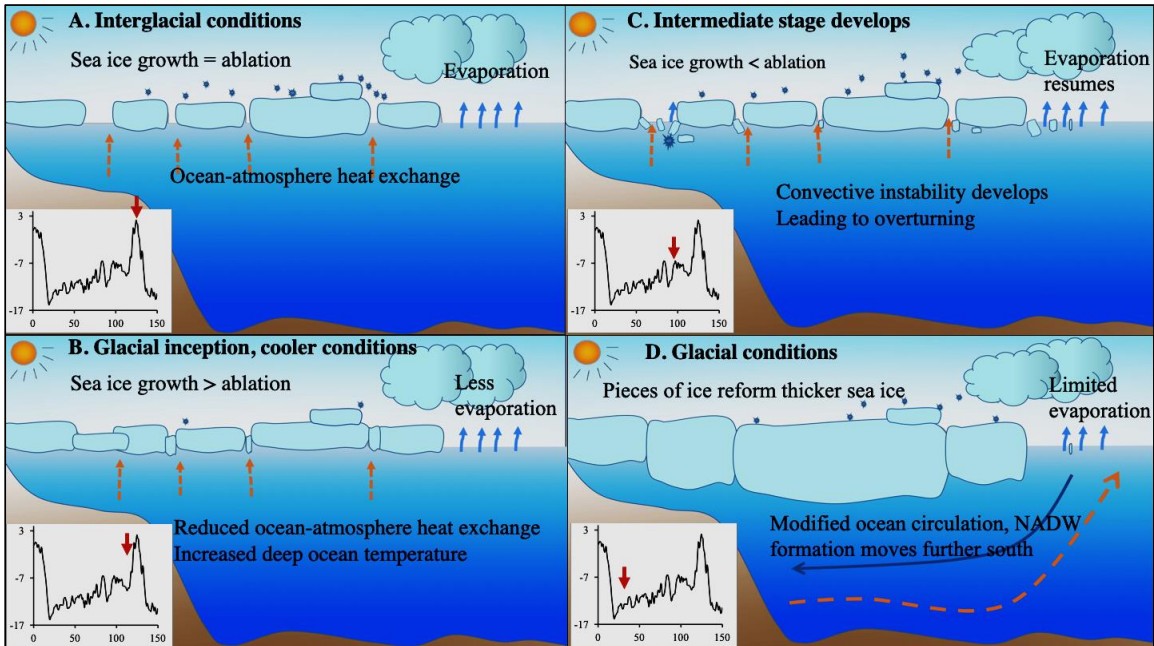

**Figure 1.** Illustration showing how the proposed two dominant negative feedbacks operate to
slow down the glaciation process with an inset showing temperature change relative to present
day north of ~ 45°N in (°C) (Bintanja and van de Wal, 2008) spanning the last 150ka. The red
arrow shows the glacial transition stage described in the given panel. **A**. Interglacial conditions
exist, there is strong exchange of energy (dashed orange arrow) and evaporation of moisture (red
arrow) from the ocean. **B**. Initiation of glacial inception and sea ice extends, insulating the ocean
while energy accumulates, increasing deep ocean temperatures and buoyancy, creating an
unstable water column. **C**. At some critical point, when the water column is sufficiently unstable
resulting in overturning and ablation of the young sea ice, albedo decreases and temperatures
increases, creating the observed intermediary stage. **D**. Sea ice has reformed from the broken
pieces of ice due to continued low summer insolation. This sea ice is thicker and sturdier, better
able to resist ocean turnover, leading to changes in the ocean circulation patterns instead.