# Peer review of "Why are glaciations slower than deglaciations?"

_Earth System Dynamics, 2019_

## Short Comment (SC1) · 19 Mar 2019

Ocean negative feedback may shorten glacial cycles instead of lengthening them

The authors suggest that ocean negative feedback may be responsible for a slow ice growth during typical ice-age cycle because "it seems intuitive that negative feedbacks would play critical roles in slowing the pace of a transition between equilibrium states".

Unfortunately, this idea is questionable. In fact, an analysis of non-linear dynamical system model of ice ages can provide a quite nuanced picture, with episodes of stability alternating with episodes of acceleration. Hence, if the mechanism suggested by the authors is real (quantitative arguments have not been provided), one could counter-argue that the cause of the asymmetry is to be found in "positive feedbacks" acting

during the deglaciation - this might actually be the most frequent explanation. The structure of ice ages cycle emerges from this interplay between negative and positive feedbacks all along the ice age cycle, so it is unclear why the asymmetry (well discussed) would necessarily point to a mechanism that has not been discussed in, to use the authors' words, the "plethora" of other models.

Contrary to author's assumption, the ice-age dynamics may be very counterintuitive. For example, it has been shown (Verbitsky et al., 2018), that regardless of the physical interpretation of the positive or negative ocean feedback, the period of glacial rhythmicity is defined by the ratio of intensities of ocean resultant (more positive or less positive) feedback to ice-sheet own negative feedback. When this ratio is high enough, the model exhibits late-Pleistocene type of rhythmicity with a period of about 100 ky. When the ratio is small (this is the case advocated by the authors, i.e., negative ocean feedback is strong enough to compensate for ocean positive feedback) the model exhibits early-Pleistocene type of fluctuations with a period of 40 ky.

Thus intensive ocean negative feedback may shorten glacial cycles instead of lengthening them.

References:

Verbitsky, M. Y., Crucifix, M., and Volobuev, D. M.: A theory of Pleistocene glacial rhythmicity, Earth Syst. Dynam., 9, 1025-1043, https://doi.org/10.5194/esd-9-1025-2018, 2018

---

## Referee Comment (RC1) · Anonymous Referee #1 · 22 Apr 2019

Overall:

This paper proposes two possible feedbacks due to Arctic sea ice that could influence Northern Hemispheric glacial-interglacial variations, in particular the sawtooth asymmetry long noted in 100,000 year cycles. Sea-ice feedbacks could well play a role in aspects of glacial-interglacial cycles, whether as primary feedbacks or as secondary influences among others. However, in my view the description of the mechanisms here is confusing, and I have several concerns that question their viability relating to the asymmetry. Given the main concerns (#1 to 4 described below), I think that the mechanisms in the paper are not organized in a physically coherent way, and the proposed sequence is not sufficiently developed and plausible for an ESD Ideas paper.

Specific points:

[Figure]

1. The text and Fig. 1 are confusing regarding the sequence of processes, mechanisms, and stages, and how they relate to the "inception" and "termination" periods of the title. The text should define the intervals (in years before present, ka) the intervals of "inception" (and also termination, presumably ∼20 to 10 ka), and the text and red arrows in the insets should specify more precisely to what parts of the long-term record the proposed mechanisms and stages apply. The time periods for each stage in the panels in Fig. 1 should be specified explicitly, either by giving ka values or with time bars in the insets relating to the long-term record.

In particular the time period referred to as "inception" is unclear. "Inception" is often taken to mean the first rapid buildup of ice following the last interglacial, around ∼120 to 110 ka. Given that, the long-term record in Fig. 1. insets does not have a pronounced difference in the slope (rate of growth), compared to the rate of retreat during the last termination ∼20 to 10 ka. By eye at least, they both look equally steep. In that case the whole premise of the paper is on shaky ground. But perhaps "inception" here refers to the later more gradual buildup over a longer part of the cycle, averaged over several orbital variations, even as long as ∼100 to 20 ka, which does give rise to the long-noted sawtooth asymmetry. This may be the case, as suggested by the phrase "changeover to a glacial period takes tens of thousands of years" (line 23).

Other sources of confusion are: (i) none of the arrows in Fig. 1, or the intervals referred to in the text, lie in the interval roughly ∼20 to 10 ka usually referred to as "termination", i.e., the last deglacial retreat since LGM, and (ii) it does not seem to matter whether the red arrows in the insets fall in a zenith or nadir of faster orbital/millennial fluctuations of the long-term record.

The long-term record in the insets of Fig. 1 is presumably from Fig. 1b of Bintanja et al. (2008), which is a model-dependent reconstruction. It would be better to use a purely data-based record, with ice-core, deep-sea-core d18O, or sea-level proxies.

2. Several aspects of the mechanisms discussed in the middle paragraphs (lines

38-89) are confusing and/or questionable. The first of the two feedbacks, "sea ice-precipitation", is reasonable and has been involved in previous studies, i.e., greater Arctic sea ice cover reduces evaporation and hence high-latitude precipitation, reducing the surface budget of Northern Hemispheric ice sheets. The second feedback, "sea ice-insulation", involves the sea-ice "lid" insulating the Arctic Ocean thermally from the atmosphere, but it is not clear in what sense this is a feedback, positive or negative, and how it influences ice-sheet growth or retreat. (It must have an effect on the latter, given the premise of the paper). Various processes are mentioned in lines 66-89 including buildup of geothermal heat flux, but how they are negative feedbacks on terrestrial ice sheet volume variations is absent or unclear.

3. Arctic sea ice, up to a few meters thick, is a "fast" component of the climate system, coming into quasi-equilibrium with the regional atmospheric and oceanic climate within a few decades, i.e, it has only decadal-scale inertia, and its mass turns over every few years to ∼decade. As such it can influence climate sensitivity to external changes (like water-vapor feedback for instance), or influence tipping points between multiple stable states. But it is not itself a long-term component of the climate system with inertial time scales of hundreds to many thousands of years (such as ice-sheet size, deep-ocean temperatures, bedrock deformation state, $CO_2$ level). This important distinction seems to be blurred in places, and sea ice is implied to have inertia of thousands of years, e.g., lines 85-86 ("since not all first-stage ice was lost") seem to require that the same sea-ice mass persists between the stages discussed here (Stage A to D, or perhaps C to D), tens of thousands of years apart. This contributes to the confusion regarding the sequence of mechanisms and processes in the middle paragraphs.

4. In places, the text neglects associated processes that are likely to dominate the process under discussion. For instance, in line 51, less summer melting of sea ice could well dominate over less precipitation, favoring and not hindering sea-ice growth. (A related point: the dominant control of Northern Hemispheric ice-sheet variations on orbital time scales is generally considered to be not snowfall, but ice melt in southern

ablation zones, i.e., summer atmospheric temperatures, not precipitation rates, as recognized by Milankovitch and many subsequent studies by choosing summer-season insolation at ∼60 or 70 N as the orbital metric). Sea ice-precipitation feedback may still function, but why it is not dominated by changes in summer air temperatures and ablation-zone melt should be justified if possible.

Another example is in lines 69-70 discussing the buildup of geothermal heat flux. The effect of this on Arctic ocean temperatures would likely be minor compared to changes in ocean circulation exchange between the deep Arctic and the North Atlantic.

5. Tziperman and Gildor (2003) referenced here and their related papers involve many of the same mechanisms as here: sea-ice switches, deep ocean temperatures, and the sea ice-precipitation feedback on ice sheets. How are the feedbacks and the sequences here different from theirs?

6. The scenarios here do not consider the possibility of very thick ∼1 km ice-shelf cover over the entire Arctic Ocean during some past glacial maxima, proposed by Jakobbson et al. (2016) referenced here. These thick ice shelves would have been supplied primarily by ice-sheet flow and would introduce very different physics and processes than here. This could at least be mentioned, as the Jakobbson study is used as a reference.

---

## Referee Comment (RC2) · Anonymous Referee #2 · 29 Apr 2019

Understanding the mechanisms of Quaternary glacial cycles remains one of the main challenges in the field of theoretical climatology and not surprisingly this problem attracts significant attention. However, although a comprehensive theory of Quaternary climate dynamics is still missing, significant progress has been achieved in recent years in the understanding of glacial cycles and numerous papers on this subject have been published. This is why proposing a new idea in this field requires good knowledge of the previous works. Unfortunately, the manuscript by Ramadhin and Yi shows that this is not the case and already the title of the manuscript and the first paragraph contain a number of factual errors.

First, the question posed in the title - "Why are glacial inceptions slower than terminations" - is not correct. Glacial inceptions have the same time scale (order of 10,000

years) as glacial terminations. The asymmetry of late Quaternary glacial cycles is manifested in the fact that typically the time intervals between the end of previous interglacials and the glacial maxima (which is not the same as glacial inception) are an order of magnitude longer than the time between glacial maxima and the onset of the next interglacial state (glacial termination).

Second, 100 kyr cyclicity dominated only over the last million years and not the entire Quaternary as the authors wrote. Prior to 1 million years ago, the dominant cyclicity of glacial cycles was 40,000 years.

Third, glacial terminations do not occur "after a maximum in NH summer insolation is crossed". Glacial terminations typically begin well before the maximum of insolation is reached and, for example, in the case of the penultimate termination, the termination has been completed several thousand years before the maximum of NH summer insolation has been crossed.

However, my main problem with the manuscript is related to the fact that the authors discuss only sea ice and not continental ice sheets. The latter is only mentioned once while "sea ice" is mentioned 30 times. This makes an odd impression that under glacial cycles the authors understand growing and retreat of sea ice rather than waning and waxing of NH continental ice sheet, which of course is incorrect. There is no doubt that sea ice in both hemispheres does play an important role in climate system dynamics and there are a number of both positive and negative feedbacks related to sea ice which are important for glacial cycles. However, the asymmetry of glacial cycles is related to continental, primarily NH, ice sheets. Whatever the role of sea ice is, it simply has too short time scale to determine durations of glacial inceptions and terminations.

---

## Author Comment (AC1) · 1 Jun 2019

We thank you for your extensive and constructive comments that are helpful to improve our manuscript. We agree with your comments and have worked diligently to address these and revised the manuscript and hopefully have improved it significantly.

Main changes are: i. Intervals referred to in the manuscript are defined in terms of Ka with arrows added to the inset of Figure 1A to clarify glacial terminations and glaciations. ii. The insets of Figure 1 has been modified to use a more data-based record of the glaciation transition. iii. The text has been revised to clarify the point that sea ice feedback mechanisms are one of many. However, what we are trying to propose in this manuscript is that negative sea ice feedbacks can play a critical role at bifurcation

points and in this way regulate the pace of the glaciation.

Our point-to-point responses to each comment are below:

Referee #1 Overall: This paper proposes two possible feedbacks due to Arctic sea ice that could influence Northern Hemispheric glacial-interglacial variations, in particular the sawtooth asymmetry long noted in 100,000 year cycles. Sea-ice feedbacks could well play a role in aspects of glacial-interglacial cycles, whether as primary feedbacks or as secondary influences among others. However, in my view the description of the mechanisms here is confusing, and I have several concerns that question their viability relating to the asymmetry. Given the main concerns (#1 to 4 described below), I think that the mechanisms in the paper are not organized in a physically coherent way, and the proposed sequence is not sufficiently developed and plausible for an ESD Ideas paper.

Authors We are grateful to you for reading and letting us know your concerns with the manuscript. We appreciate the details in your comments, and it has served us well in helping us improve our understanding of some concepts and it is appreciated, thanks. Referee #1 Specific points: 1. The text and Fig. 1 are confusing regarding the sequence of processes, mechanisms, and stages, and how they relate to the "inception" and "termination" periods of the title. The text should define the intervals (in years before present, ka) the intervals of "inception" (and also termination, presumably âĹij20 to 10 ka), and the text and red arrows in the insets should specify more precisely to what parts of the long-term record the proposed mechanisms and stages apply. The time periods for each stage in the panels in Fig. 1 should be specified explicitly, either by giving ka values or with time bars in the insets relating to the long-term record.

Authors You raise a good point, we have revised the manuscript to reflect this by defining glacial inception, termination, intermediate stage and the full transition to glacial conditions in terms of ka for the last termination and glaciation. In doing this, we realized the term 'glacial inception' may not be the best words to describe what we meant

and have replaced it with 'glaciation'.

Authors changes in manuscript Therefore, the title of the paper now reads "Why are glaciations slower than deglaciations?"

Referee #1 In particular the time period referred to as "inception" is unclear. "Inception" is often taken to mean the first rapid buildup of ice following the last interglacial, around âĹij120 to 110 ka. Given that, the long-term record in Fig. 1. insets does not have a pronounced difference in the slope (rate of growth), compared to the rate of retreat during the last termination âĹij20 to 10 ka. By eye at least, they both look equally steep. In that case the whole premise of the paper is on shaky ground. But perhaps "inception" here refers to the later more gradual buildup over a longer part of the cycle, averaged over several orbital variations, even as long as âĹij100 to 20 ka, which does give rise to the long-noted sawtooth asymmetry. This may be the case, as suggested by the phrase "changeover to a glacial period takes tens of thousands of years" (line 23).

Authors Thank you, we have now clarified this point by defining the intervals; glacial inception and the glaciation process which includes both the glacial inception and intermediate stages. By pointing this out perhaps you have helped us to significantly improve this manuscript.

Authors changes in manuscript From benthic $\delta$18O (‰ records from ODP Site 983 from Raymo et al., (2004) the duration of the last termination is $\sim$ 10ka while the glaciation process including the inception and intermediate stage had a duration of $\sim$ 77ka.

Referee #1 Other sources of confusion are: (i) none of the arrows in Fig. 1, or the intervals referred to in the text, lie in the interval roughly âĹij20 to 10 ka usually referred to as "termination", i.e., the last deglacial retreat since LGM, and (ii) it does not seem to matter whether the red arrows in the insets fall in a zenith or nadir of faster orbital/millennial fluctuations of the long-term record.

Authors To show the difference in the rates between glacial terminations and the glaciation process, gray arrows are used to indicate these in the inset of Figure 1A.

Referee #1 The long-term record in the insets of Fig. 1 is presumably from Fig. 1b of Bintanja et al. (2008), which is a model-dependent reconstruction. It would be better to use a purely data-based record, with ice-core, deep-sea-core d18O, or sea-level proxies.

Authors We have revised the insets of Figure 1 using benthic $\delta$18O (‰ records from ODP Site 983 in the North Atlantic by Raymo et al., (2004).

Referee #1 2. Several aspects of the mechanisms discussed in the middle paragraphs (lines 38-89) are confusing and/or questionable. The first of the two feedbacks, "sea ice- precipitation", is reasonable and has been involved in previous studies, i.e., greater Arctic sea ice cover reduces evaporation and hence high-latitude precipitation, reducing the surface budget of Northern Hemispheric ice sheets. The second feedback, "sea ice-insulation", involves the sea-ice "lid" insulating the Arctic Ocean thermally from the atmosphere, but it is not clear in what sense this is a feedback, positive or negative, and how it influences ice-sheet growth or retreat. (It must have an effect on the latter, given the premise of the paper). Various processes are mentioned in lines 66-89 including buildup of geothermal heat flux, but how they are negative feedbacks on terrestrial ice sheet volume variations is absent or unclear.

Authors The second feedback aids sea ice ablation and a return to decreased albedo and warmer temperatures and in this way serves as a negative feedback. We had adjusted a few lines in paragraph 4 to make this clear.

Authors changes in manuscript "Sea ice disintegration decreases albedo resulting in a return to higher temperatures and the development of the intermediary stage of almost interglacial conditions depicted in Figure 1C. The intermediary stage of reduced albedo and warmer conditions may have a wider influence by affecting other climate variables such as land-based ice-sheets which can in turn impact the behavior of other

dominant players in this climate transition." Referee #1 3. Arctic sea ice, up to a few meters thick, is a "fast" component of the climate system, coming into quasi-equilibrium with the regional atmospheric and oceanic climate within a few decades, i.e, it has only decadal-scale inertia, and its mass turns over every few years to âĹijdecade. As such it can influence climate sensitivity to external changes (like water-vapor feedback for instance), or influence tipping points between multiple stable states. But it is not itself a long-term component of the climate system with inertial time scales of hundreds to many thousands of years (such as ice-sheet size, deep-ocean temperatures, bedrock deformation state, CO2 level). This important distinction seems to be blurred in places, and sea ice is implied to have inertia of thousands of years, e.g., lines 85-86 ("since not all first-stage ice was lost") seem to require that the same sea-ice mass persists between the stages discussed here (Stage A to D, or perhaps C to D), tens of thousands of years apart. This contributes to the confusion regarding the sequence of mechanisms and processes in the middle paragraphs.

Authors We agree that these feedbacks are only temporarily dominant and their influence changes depending on how close to bifurcation points the system may be. The text has been revised to increase its clarity.

Authors changes in manuscript This stage remains dominant temporarily with sea ice cover formation increasing and ablation processes following until gradually sturdier sea ice forms as overall, temperatures cooler than interglacial temperatures prevail as shown in Figure 1D.

Referee #1 4. In places, the text neglects associated processes that are likely to dominate the process under discussion. For instance, in line 51, less summer melting of sea ice could well dominate over less precipitation, favoring and not hindering sea-ice growth. (A related point: the dominant control of Northern Hemispheric ice-sheet variations on orbital time scales is generally considered to be not snowfall, but ice melt in southern ablation zones, i.e., summer atmospheric temperatures, not precipitation rates, as recognized by Milankovitch and many subsequent studies by choosing

summer-season insolation at âĹij60 or 70 N as the orbital metric). Sea ice-precipitation feedback may still function, but why it is not dominated by changes in summer air temperatures and ablation-zone melt should be justified if possible.

Authors Gradual changes in a forcing can result in little change in a dynamic system until some critical threshold has been surpassed when a sudden large shift may be seen in the system (Scheffer et al., 2012). We think that may be the case here, the sea ice-precipitation feedback though still functional throughout the glaciation is most effective in assisting large change at critical points.

Referee #1 Another example is in lines 69-70 discussing the buildup of geothermal heat flux. The effect of this on Arctic ocean temperatures would likely be minor compared to changes in ocean circulation exchange between the deep Arctic and the North Atlantic.

Authors A previous model study (Rial and Saha, 2011) has shown that sea ice cover can cause buildup of geothermal heat flux in the deep ocean increasing temperature so it is higher than surface temperatures resulting in increased buoyancy of the deep ocean and eventually increased vertical turbulence which enhances the break up of the sea ice. While this may be minor, we think it provides enough change at the tipping point to facilitate the development of an intermediate stage and may explain why this mechanism is not dominant throughout the glacial stage.

Referee #1 5. Tziperman and Gildor (2003) referenced here and their related papers involve many of the same mechanisms as here: sea-ice switches, deep ocean temperatures, and the sea ice-precipitation feedback on ice sheets. How are the feedbacks and the sequences here different from theirs?

Authors Gildor and Tziperman, (2000) proposed a sea ice switch mechanism which says the sea ice acts as a control of the atmospheric moisture fluxes and precipitation through its albedo and insulating effects switching it between two modes: growing land ice and retreating land ice. In the current proposed mechanism, the sea ice is also thought to control atmospheric moisture and precipitation. However, the mechanism

presented here differs in that it considers the effect of insulation on the temperature and stability of the deep ocean instead of land ice sheets. Here sea ice cover is considered as a control on deep ocean temperature in the Arctic which in turn can control the extent of sea ice cover by vertical turbulence. Authors changes in manuscript Gildor and Tziperman, (2000) proposed a sea ice switch mechanism which says the sea ice acts as a control of the atmospheric moisture fluxes and precipitation through its albedo and insulating effects switching it between two modes: growing land ice and retreating land ice. A similar mechanism is presented here but differs in that it considers the effect of sea ice insulation on the temperature and stability of the deep ocean instead of land ice sheets. Referee #1 6. The scenarios here do not consider the possibility of very thick âĹij1 km ice-shelf cover over the entire Arctic Ocean during some past glacial maxima, proposed by Jakobbson et al. (2016) referenced here. These thick ice shelves would have been supplied primarily by ice-sheet flow and would introduce very different physics and processes than here. This could at least be mentioned, as the Jakobbson study is used as a reference.

Authors There is indeed a possibility of the Arctic Ocean being covered by very thick sea ice which we think follows the intermediate stage developed during the glaciation transition process. We agree that the development of a thick $\sim$ 1km ice cover over the Arctic has a high possibility and can lead to various other processes not considered in our proposal. However, the proposed mechanism in this paper and its resultant processes is thought to be dominant only during the transition where the extent of sea ice is most important.

Please also note the supplement to this comment:
https://www.earth-syst-dynam-discuss.net/esd-2019-10/esd-2019-10-AC1-supplement.pdf

---

## Author Comment (AC3) · 2 Jun 2019

Referee #2 Understanding the mechanisms of Quaternary glacial cycles remains one of the main challenges in the field of theoretical climatology and not surprisingly this problem attracts significant attention. However, although a comprehensive theory of Quaternary climate dynamics is still missing, significant progress has been achieved in recent years in the understanding of glacial cycles and numerous papers on this subject have been published. This is why proposing a new idea in this field requires good knowledge of the previous works. Unfortunately, the manuscript by Ramadhin and Yi shows that this is not the case and already the title of the manuscript and the first paragraph contain a number of factual errors.

Authors Thank you for reading our manuscript and providing constructive comments which have helped to improve the manuscript. We have worked diligently to address each comments and revised the manuscript accordingly.

Referee #2 First, the question posed in the title - "Why are glacial inceptions slower than terminations" - is not correct. Glacial inceptions have the same time scale (order of 10,000 years) as glacial terminations. The asymmetry of late Quaternary glacial cycles is manifested in the fact that typically the time intervals between the end of previous interglacials and the glacial maxima (which is not the same as glacial inception) are an order of magnitude longer than the time between glacial maxima and the onset of the next interglacial state (glacial termination).

Authors Thanks for your insightful comment. We agree and have modified the title to "Why are glaciations slower than deglaciations?". Additionally, we have defined the intervals referred to in the paper; glacial inceptions, glacial terminations and glaciation in terms of number of years from present day.

Authors changes in manuscript From benthic $\delta18O$ (‰ records from ODP Site 983 from Raymo et al., (2004) the duration of the last termination is $\sim$ 10ka while the glaciation process including the inception and intermediate stage had a duration of $\sim$ 77ka.

We think this allows sea ice to extend rapidly, increasing albedo, further decreasing temperatures and helps to explain the initial rapid temperature drop observed for this glacial inception period which lasts $\sim$10ka.

Based on temperature reconstruction, the intermediate stage has a duration of $\sim$42ka, where there is increased evaporation due to greater exposed ocean surfaces relative to the initial glacial inception period, and increased precipitation.

Referee #2 Second, 100 kyr cyclicity dominated only over the last million years and not the entire Quaternary as the authors wrote. Prior to 1 million years ago, the dominant cyclicity of glacial cycles was 40,000 years.

Authors This is true, we have revised the manuscript to clarify the period we are referring to which is the latter part of the Pleistocene glaciations, after the MPT.

Authors changes in manuscript Paleoclimate data show that the Earth's climate of the last 2.6Ma is dominated by cold glaciations, recently (after the Mid Pleistocene Transition) the duration of these ice ages is ∼100ka with extensive glaciers and warm interglacials with little global ice cover lasting 10-30ka (Imbrie et al., 1992).

Referee #2 Third, glacial terminations do not occur "after a maximum in NH summer insolation is crossed". Glacial terminations typically begin well before the maximum of insolation is reached and, for example, in the case of the penultimate termination, the termination has been completed several thousand years before the maximum of NH summer insolation has been crossed.

Authors We have modified the manuscript to reflect this point, thank you.

Authors changes in manuscript The termination of a glacial period occurs rapidly while the changeover to a glacial period takes tens of thousands of years to be completed resulting in an interesting asymmetrical shape for which there is yet no consensus on the mechanism(s) (Tziperman and Gildor, 2003).

Referee #2 However, my main problem with the manuscript is related to the fact that the authors discuss only sea ice and not continental ice sheets. The latter is only mentioned once while "sea ice" is mentioned 30 times. This makes an odd impression that under glacial cycles the authors understand growing and retreat of sea ice rather than waning and waxing of NH continental ice sheet, which of course is incorrect. There is no doubt that sea ice in both hemispheres does play an important role in climate system dynamics and there are a number of both positive and negative feedbacks related to sea ice which are important for glacial cycles. However, the asymmetry of glacial cycles is related to continental, primarily NH, ice sheets. Whatever the role of sea ice is, it simply has too short time scale to determine durations of glacial inceptions and terminations.

Authors Thank you for pointing this out, relative to land based ice sheets, sea ice is short lived. However, here we think the role of sea though short lived and relatively minor plays an important role when the system is at a bifurcation point. It may serve as a critical parameter in helping the changeover between stable states. Though these sea ice feedbacks may be active during all stages of glacial-interglacial climate variations, they may be critical after thresholds have been surpassed.

Please also note the supplement to this comment:
https://www.earth-syst-dynam-discuss.net/esd-2019-10/esd-2019-10-AC3-supplement.pdf

---

## Author Comment (AC4) · 2 Jun 2019

Short comments (SCs) from the scientific community
Ocean negative feedback may shorten glacial cycles instead of lengthening them The authors suggest that ocean negative feedback may be responsible for a slow ice growth during typical ice-age cycle because "it seems intuitive that negative feedbacks would play critical roles in slowing the pace of a transition between equilibrium states". Unfortunately, this idea is questionable. In fact, an analysis of non-linear dynamical system

model of ice ages can provide a quite nuanced picture, with episodes of stability alternating with episodes of acceleration. Hence, if the mechanism suggested by the authors is real (quantitative arguments have not been provided), one could counterargue that the cause of the asymmetry is to be found in "positive feedbacks" acting during the deglaciation - this might actually be the most frequent explanation. The structure of ice ages cycle emerges from this interplay between negative and positive feedbacks all along the ice age cycle, so it is unclear why the asymmetry (well discussed) would necessarily point to a mechanism that has not been discussed in, to use the authors' words, the "plethora" of other models. Contrary to author's assumption, the ice-age dynamics may be very counterintuitive. For example, it has been shown (Verbitsky et al., 2018), that regardless of the physical interpretation of the positive or negative ocean feedback, the period of glacial rhythmicity is defined by the ratio of intensities of ocean resultant (more positive or less positive) feedback to ice-sheet own negative feedback. When this ratio is high enough, the model exhibits late-Pleistocene type of rhythmicity with a period of about 100 ky. When the ratio is small (this is the case advocated by the authors, i.e., negative ocean feedback is strong enough to compensate for ocean positive feedback) the model exhibits early-Pleistocene type of fluctuations with a period of 40 ky. Thus intensive ocean negative feedback may shorten glacial cycles instead of lengthening them.

References: Verbitsky, M. Y., Crucifix, M., and Volobuev, D. M.: A theory of Pleistocene glacial rhythmicity, Earth Syst. Dynam., 9, 1025-1043, https://doi.org/10.5194/esd-9-1025- 2018, 2018 Interactive comment on Earth Syst. Dynam. Discuss., https://doi.org/10.5194/esd-2019-10, 2019.

Authors Response

Dear Dr. Mikhail Verbitsky,

We are very thankful for the insightful comment.

Verbitsky et al., (2018) presents a model that introduces a dimensionless variability

factor, V. This factor, V is large ( 0.75) when positive feedbacks dominate, producing 100 kyr cycles while when less positive, V is small ( 0) and 40 kyr cycles dominate. Since we proposed the dominance of negative feedbacks to explain the slower glacial inception process versus the faster glacial termination, then V should be small in our case and produce 40 kyr cycles instead of 100 kyr observed for the late Pleistocene.

We think in this scenario, yes this seems to be true. However, in this proposal, the conditions facilitating the dominance of the negative sea ice feedbacks are short-lived, and the proposed feedbacks do not dominate for the entirety of the glaciation. If it did then the intermediate stage would be longer and yes the glacial cycle shorter. However, as conditions change to become less favorable, these feedbacks progressively becomes less dominant. For example, ocean circulation changes in the North Atlantic, as the temperature drops, North Atlantic Deep Water formation moves further south as demonstrated by Rahmstorf (2006).

What we are proposing in relation to the V factor in Verbitsky et al., (2018) model means the V may be variable. V is large at the initiation of glacial inception (positive feedbacks are strong) and smaller during the intermediate stage (positive feedbacks are less strong) during the inception process. However, as conditions change the V factor gets larger and the transition to glacial conditions continue leaving the intermediate stage behind.

References

Rahmstorf, S.: Thermohaline Ocean Circulation. In: Encyclopedia of Quaternary Sciences, Edited by S. A. Elias Elsevier Amsterdam, 2006.

Verbitsky, M. Y., Crucifix, M., and Volobuev, D. M.: A theory of Pleistocene glacial rhythmicity, Earth Syst. Dynam., 9, 1025-1043, https://doi.org/10.5194/esd-9-1025-2018, 2018

Please also note the supplement to this comment:

https://www.earth-syst-dynam-discuss.net/esd-2019-10/esd-2019-10-AC4-supplement.pdf